# SHAPING LATENT DIFFUSION FOR EFFICIENT TEXT-CONDITIONED INTERACTION GENERATION

## ABSTRACT

Existing latent diffusion models excel at text-to-motion generation for single-person, but struggle with multi-person scenarios. This is largely due to the limited capacity of the latent representation, which fails to capture complex spatio-temporal dynamics between individuals (*e.g.*, relative orientation). To address this, we introduce **Interaction Latent Diffusion (ILD)** model. Unlike previous methods using the single-token latent space under geometric constraint, ILD leverages an interaction-aware, multi-token latent space that is enhanced by inter-person constraints and aligned with pretrained tokenizers, strengthening its expressibility. Building on ILD, we further improve the physical plausibility and ensure real-time inference by introducing two key components. Firstly, we propose an efficient neural collision guidance combined with high-order ODE solvers, avoiding the costly occupancy-based detection while reducing artifacts and latency. Secondly, we develop Flash ILD (FILD), a distilled model capable of one-step generation through a tailored consistency distillation and distribution matching pipeline. We evaluate the proposed ILD and FILD qualitatively and quantitatively on InterHuman and Inter-X datasets. Specifically, on the InterHuman dataset, ILD achieves a new state-of-the-art FID of 4.869 (vs. 5.154 for Inter-Mask), meanwhile FILD accelerates inference from 10 FPS to 30 FPS. The code will be available.

## 1 INTRODUCTION

As a powerful generative model, diffusion models (DMs) have been widely used in synthetic human motion generation, such as text-to-motion tasks (Tevet et al., 2022b; Liang et al., 2024; Guo et al., 2024; Zhang et al., 2023; Tevet et al., 2022a; Cai et al., 2024) generating high-quality and diverse motions by effectively modeling many-to-many distributions. However, their success in single-person synthesis has not translated to multi-person interactions. They suffer from intensive computational costs, a lack of physical grounding, and, most critically, a representational bottleneck. These limitations hinder applications in downstream tasks, such as robotics, which require quick response time (*e.g.*, 30 FPS (Goyal et al., 2024)) and physically plausible movement.

As shown in Fig. 1, previous multi-person generation methods (Ponce et al., 2024; Liang et al., 2024) use two-stream diffusion models with shared weights for each person, struggling with sampling efficiency. In contrast, existing latent diffusion (Rombach et al., 2022) models have demonstrated remarkable inference time reduction and high-fidelity generation for single-person tasks (Dai et al., 2024; Zhu et al., 2025), which use Variational Auto-Encoders (VAEs) (Kingma, 2013) to compress motion data before conditional generation via diffusion models (Ho et al., 2020). However, generalizing latent diffusion for multi-

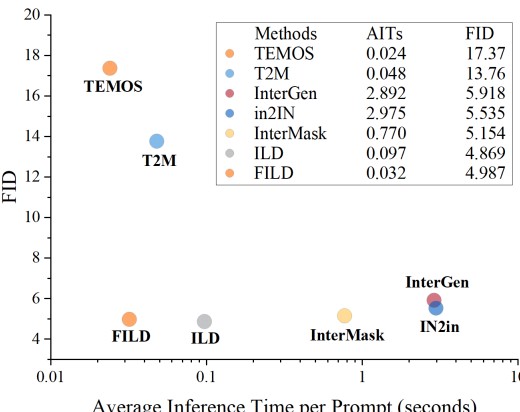

Figure 1: The motion generation quality (FID score) and speed (AITS) comparisons between ILD, FILD, and SOTA methods on the Interhuman dataset. The model closer to the origin is better. All tests are performed on one NVIDIA A100.

person scenarios remains challenging (Li et al., 2024a): 1) It is difficult to model spatial-temporal relationships between individuals, such as relative orientation (Li et al., 2024b). Because single-token latent spaces (*i.e.* $1 \times 256$) (Chen et al., 2023) lack such expressive capacity. 2) Multi-token spaces theoretically offer higher reconstruction fidelity, but diffusion models often fail to learn the complex coordination between tokens (Xie et al., 2024b; Hansen-Estruch et al., 2025).

To address this, we propose Interaction Latent Diffusion (ILD), a generative model that jointly captures multi-person motion dynamics. Our key insight is to design a unibranch VAE that compresses multi-person motion into an interaction-aware, multi-token latent space, simultaneously improving the VAE's reconstruction accuracy and the upper bound on diffusion generation quality. Firstly, we apply geometric and interactive constraints to capture complex inter- and intra-personal patterns. Secondly, we unlock the capabilities of multi-token latent spaces (*e.g.* $4 \times 512$) by aligning with a pretrained discrete motion prior (Guo et al., 2024). Notably, unlike previous multi-token methods (Xie et al., 2024b; Dai et al., 2025), these designs enhance the VAE capacity without significantly increasing the diffusion computational overhead.

While promising, efficiently generating physically plausible motion with diffusion models remains an open challenge, where the sampling efficiency depends on (Chadebec et al., 2024; Kohler et al., 2024) the per-step cost and the total iteration steps. To ensure physical grounding, previous methods (Li et al., 2024b; Jiang et al., 2025b) rely on complex post-hoc optimization at each sampling step, typically involving time-consuming occupancy-based collision detection (Mihajlovic et al., 2022). To reduce the per-step cost, we introduce a lightweight neural collision guidance (Mihajlovic et al., 2025) that efficiently penalizes interpenetration. Moreover, to reduce the total number of sampling steps, we replace the standard DDIM sampler with a high-order ODE solver (Zhang & Chen, 2022). Together, these training-free enhancements enable ILD to generate physically plausible interactions with competitive fidelity in a highly efficient 10-step regime.

Building on ILD's ability to generate high-quality interactions, we further develop an enhanced version called Flash Interaction Latent Diffusion (FILD) model. FILD achieves real-time, few-step inference by overcoming the numerical instability in few-step sampling (Chadebec et al., 2024; Zhou et al., 2024). Inspired by Dai et al. (2024), we distil a student network (i.e., FILD) from a teacher network (i.e., ILD) using consistency models (Song et al., 2023), while simultaneously refining the guidance training scheme to enhance performance. Rather than relying on pairwise loss, we employ distribution matching distillation (Yin et al., 2024) to align the student and teacher output distributions, which faithfully replicates the teacher's noise-to-sample mapping. By fine-tuning diffusions to learn both data distributions and 'fake' distributions produced by our distilled generator, we steer synthetic interactions towards higher realism.

In summary, our contributions are as follows.

- We propose Interaction Latent Diffusion (ILD), which features an interaction-aware VAE with novel alignment constraints. These constraints structure the multi-token latent space to unlock better diffusion training, resulting in both superior performance and reduced denoising latency.

- We enhance ILD's physical grounding and efficiency via a neural collision guidance that eliminates costly occupancy-based optimisation, and a high-order ODE solver that reduces sampling steps by $5\times$ (from 50 to 10) without compromising fidelity.

- We further improve ILD's efficiency by introducing the Flash Interaction Latent Diffusion (FILD) model. The pretrained ILD is distilled to a student denoiser via tailored consistency models and employs distribution matching to stabilize the generation quality for 1-step real-time sampling.

- Extensive experiments demonstrate ILD's superior generation fidelity, achieving SoTA FID score on InterHuman (4.869 vs. InterMask's 5.154) and Inter-X (0.297 vs. InterMask's 0.399) datasets. Concurrently, its real-time counterpart (i.e., 30 FPS), FILD, achieves a competitive FID score of 4.980 on InterHuman.

## 2 RELATED WORK

### 2.1 TEXT-CONDITIONED MOTION GENERATION

Given the stochastic and diverse nature of human motion, denoising Diffusion Probabilistic Models (DDPM) have been one of the most dominant methods in the area of motion generation (Yuan et al., 2023; Alexanderson et al., 2023; Barquero et al., 2023; Zhang et al., 2024b; Hoang et al., 2024). Early work by Zhang *et al.* applied diffusion models to text-conditioned human motion generation with MotionDiffuse (Zhang et al., 2024a), enhancing the diversity and plausibility of gener-

ated motions. Subsequently, Guy *et al.*introduced an adapted classifier-free diffusion-based model, Motion Diffusion Model (MDM) (Tevet et al., 2022b), to denoise the signal by predicting original motions instead of added noise. Similar to us, MDM requires fewer GPU resources and can further improve performance by utilizing extra geometric losses. For interaction motion generation, Shafir *et al.*introduced ComMDM (Shafir et al., 2023), which uses two pretrained MDM priors and a communication block to coordinate interactions between motions. Then, Tanaka *et al.*proposed the Role-aware Interaction Generation Diffusion-based model (Tanaka & Fujiwara, 2023), which incorporates semantic information such as active and passive roles from the action labels. More recently, Ponce *et al.* (Ponce et al., 2024) proposed in2IN to capture the intra-personal dynamics, conditioned not only on the overall interaction but also on the extra individual descriptions from each person. Concurrently, InterMask (Guo et al., 2024) utilizes 2D Vector Quantised VAE (Van Den Oord et al., 2017) to convert motion sequences into discrete token maps under geometric constraint and a generative masked modeling (He et al., 2022) framework to reconstruct it. However, it ignores inter-individual constraints in token maps, and its two-stream individual encoder design may hinder inference speed. In contrast, under a joint distribution of two individuals, we present ILD to capture additional interactive information in the latent space.

## 2.2 Fast Text-to-Motion Diffusion Models

The mode-covering behaviour of diffusion models makes them prone to spending excessive amounts of capacity for capturing imperceptible details of the data and thus requires huge computing resources and long inference times (Chen et al., 2023). To tackle this, Chen *et al.*introduced latent diffusion models (Rombach et al., 2022) to significantly improve both the training and sampling efficiency of denoising diffusion models without degrading their quality. Dai *et al.*proposed a Motion Latent Consistency Model (Dai et al., 2024) via latent consistency distillation, extending controllable motion generation to a millisecond level. Recently, MotionPCM (Jiang et al., 2025a) introduced the multi-interval design of Phased Consistency Model, reducing accumulated random noise in multi-step sampling. However, those methods are tailored for single-person motion generation, and the applied single-token results in artifacts for interaction generation Our work shows that ILD can reduce per-step evaluation latency and FILD can further conduct the distillation learning for the one-step generation. To enhance the real-time performance, FILD incorporates distribution matching constraints (Yin et al., 2024) to stabilise distillation learning via the consistency models. Finally, the Human-X (Ji et al., 2025) framework achieves real-time reaction (i.e., single-agent-motion) generation by coupling a low-frequency diffusion planner with a high-frequency physics tracker. Instead, our work improves the intrinsic efficiency of diffusion models for dual-agent generation.

## 3 Method

Given a text prompt, our goal is to generate an interaction sequence $x_I^{1:N} = \{x_a, x_b\}^{1:N} \in \mathbb{R}^{N \times 2D}$, where $x_a \in \mathbb{R}^{N \times D}$ and $x_b \in \mathbb{R}^{N \times D}$ represent the motion sequences of individual participants (see details in Sec 4.1). Here, $N$ and $D$ denote the length and dimensionality, respectively.

### 3.1 Interaction-Aware Variational Auto-Encoder

Our Interaction-Aware VAE (IA-VAE), illustrated in **Stage 1** of Fig. 2, learns a continuous latent space for the entire interaction sequence with variational inference. The interaction transformer encoder and decoder consist of the transformer encoder and decoder, with skip connections and layer norms, respectively. Similar to TEMOS (Petrovich et al., 2022), the interaction encoder takes a sequence of interaction of arbitrary length $x_I^{1:N} = \{x_a, x_b\}^{1:N} \in \mathbb{R}^{N \times 2D}$ as input and compresses $x^{1:N}$ into a latent representation $z \in \mathbb{R}^{L \times K}$ with high informative density. Then, the interaction decoder reconstructs the latent vector $z$ into motion sequences $\hat{x}_I^{1:N} = \{\hat{x}_a, \hat{x}_b\}^{1:N}$.

**Interaction-aware Latent Space**. In a typical VAE training process, motion reconstruction $x^{1:N}$ is constrained by the Mean Squared Error (MSE) and Kullback-Leibler (KL) losses. However, motion reconstruction generally requires more regularization for better fidelity, such as the commonly used geometric loss (Tevet et al., 2022b), which prevents intra-person artifacts from generating smooth and natural motions. Thus, following Javed et al. (2024); Li et al. (2024a), we applied the Bone length (BL) loss and the foot contacting loss as the geometric loss of IA-VAE:

$$\mathcal{L}_{\text{geometric}} = \lambda_{\text{BL}} \, \mathcal{L}_{\text{BL}} + \lambda_{\text{FC}} \, \mathcal{L}_{\text{foot}} . \tag{1}$$

In practice, these constraints are insufficient to train a robust representation for interaction reconstruction, likely due to the high uncertainty from the interplay of two people. To handle the complex spatial-temporal relationships between individuals, especially for the relative position and orientation (Li et al., 2024b), we further introduce interactive losses(Liang et al., 2024) for IA-VAE training,

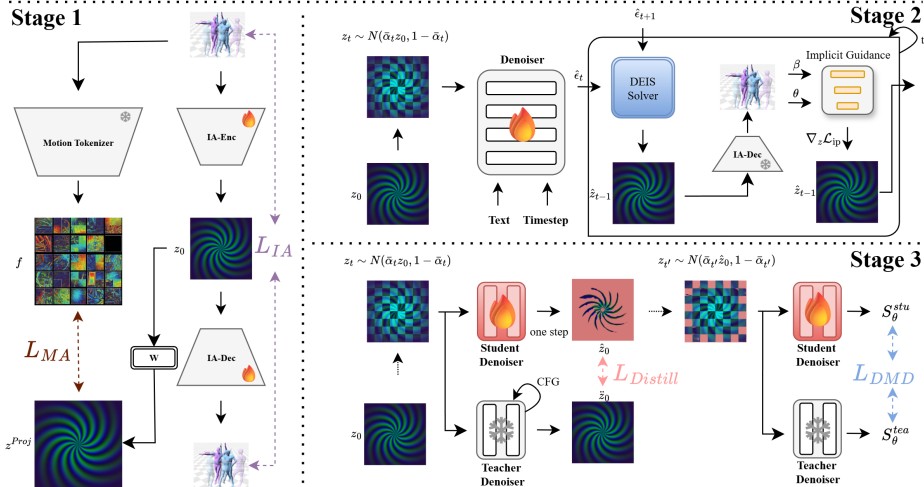

Figure 2: Overview of the proposed pipeline. **Stage 1**: An Interaction-Aware VAE (IA-VAE) is designed to learns an expressive, multi-token latent space by introducing interaction-aware constraints and motion tokeniser alignment loss (Sec.3.1). **Stage 2**: Interaction Latent Diffusion Denoiser employs a second-order DEIS solver to reduce sampling to just 10 steps, while a lightweight implicit guidance maintains physical plausibility at test-time (Sec.3.2). **Stage 3**: Flash Interaction Latent Diffusion (FILD) distils the 10-step model into a one-step generator using a tailored consistency distillation framework, further refined by distribution matching (Sec.3.3).

comprising masked joint distance map (DM) loss and relative orientation (RO) loss:

$$\mathcal{L}_{\text{interactive}} = \lambda_{\text{DM}} \, \mathcal{L}_{\text{DM}} + \lambda_{\text{RO}} \, \mathcal{L}_{\text{RO}} \,. \tag{2}$$

We follow the training objective design in Liang et al. (2024), and refer the reader to Appendix A for details. Finally, we combine them into a interaction-aware loss: $\mathcal{L}_{ia} = \mathcal{L}_{\text{geometric}} + \mathcal{L}_{\text{interactive}}$.

**Motion Tokenizer Alignment.** Beyond spatial-temporal regularization, the dimensionality of the latent space is critical (Hansen-Estruch et al., 2025). While a higher-resolution latent space improves reconstruction, it often degrades the fidelity of the subsequent diffusion process (Xie et al., 2024a), as it requires substantially larger diffusion models and more training iterations to achieve comparable generation performance. On the other hand, previous methods (Chen et al., 2023; Dai et al., 2024) ease the learning of the denoising process by simplifying the target distribution via the low-dimensional latent code (*e.g.* $1 \times 256$), which unavoidably bottlenecks the generative model, as the auto-encoder struggles with effective motion reconstruction.

To resolve this dimensionality trade-off, we introduce a novel motion alignment loss (Yao et al., 2025) (See Eq. 3), which forces the latent space of IA-VAE to align with a powerful, pretrained discrete tokenizer (*i.e.*, Vector Quantised-VAE (VQ-VAE)). Unlike Xie et al. (2024b) using cascade diffusion to scale up model parameters, we enhance the capacity of the multi-token latent space, as well as diffusion performance, without increasing diffusion parameters or significant extra computational cost (see Table 2d).

This approach is motivated by the key insight that discrete VQ-VAEs can learn more robust and efficient motion representations compared to their continuous counterparts (Zhang et al., 2023; Javed et al., 2024), which often struggle with redundancy and error amplification (Meng et al., 2024). Therefore, instead of simplifying our latent space, we leverage a pretrained Residual VQ-VAE (RVQ-VAE) (Guo et al., 2024) as a prior tokenizer and use an alignment loss to transfer its structural properties to the multi-token latent space. Compared to previous single-pass motion VQ tokenizers (Zhang et al., 2023; Javed et al., 2024), RVQ-VAE employs iterative rounds of residual quantization to reduce quantization errors progressively (Guo et al., 2024).

Especially, due to the incompatibility between IA-VAE and RVQ-VAE dimensionality, we need to project motion latents to match them. We evaluate interpolation, downsampling, and linear layer transformation, where we find the linear layer transformation provides the best results (See Appendix B for more details). In detail, the projected latent space $z_{ij}^{Proj}$ is forced to align with the

well-structured space $f_{ij}$ of the pretrained RQ-VAE tokenizer, which minimizes the cosine similarity gap with a margin $m_1$ at each spatial location $(i, j)$:

$$\mathcal{L}_{\text{ma}} = \frac{1}{h \times w} \sum_{i=1}^{h} \sum_{j=1}^{w} \text{ReLU} \left( 1 - m_1 - \frac{z_{ij}^{Proj} \cdot f_{ij}}{\|z_{ij}^{Proj}\| \|f_{ij}\|} \right). \tag{3}$$

**VAE Training Loss**. This comprehensive training objective guides the IA-VAE to learn a high-resolution latent space that respects both geometric and interactive constraints while aligning the discrete motion tokenizer prior: $\mathcal{L}_{\text{IA-VAE}} = \lambda_{\text{ma}} * \mathcal{L}_{\text{ma}} + \lambda_{\text{ia}} * \mathcal{L}_{\text{ia}} + \lambda_{\text{KL}} \mathcal{L}_{\text{KL}} + \mathcal{L}_{\text{rec}}$.

### 3.2 INTERACTION LATENT DIFFUSION DENOISER

As shown in **Stage 2** of Fig. 2, ILD denoiser $\varepsilon_\theta$ takes as input a latent token $z_0$ output by the encoder of IA-VAE. The denoiser is designed to iteratively anneal the noise from a Gaussian distribution to a latent space distribution $p(z)$, by learning the noise prediction from a Markov process, giving $\{z_t\}_{t=1}^{T}$. Compared to raw interaction sequences $x_I^{1:N}$, the interaction-aware latent token $z_0$ enhances the performance by removing high-frequency outliers and accelerating convergence. We use the skip-transformers as the backbone of the denoiser, and apply the in-context condition mechanism. Diffusion models consist of two interconnected processes, namely, forward and backward.

**Multi-token Latent Computational Overhead**. A potential concern is that the multi-token latent space may significantly increase diffusion computational overhead due to the larger input. We proposed to alleviate this by applying in-context learning (Ju et al., 2025), rather than the adaptive condition mechanism (Peebles & Xie, 2023). In detail, time embedding $t$ and text embedding $c$ are first added and then concatenated with latent embedding $z$, thus the input dimension $d_{in} = d_t + d_c + d_z$, rather than $d'_{in} = d_z$ for its adaptive counterpart. In practice, under a certain number (*e.g.*, $4 \times 512$), the larger token length results in a negligible increase in the inference time (See Tab. 2d).

**Forward Process**. The forward diffusion process gradually corrupts the data by interpolating between a sampled data point $\mathbf{z}_0$ and Gaussian noise $\epsilon \sim \mathcal{N}(0, \mathbf{I})$. That is:

$$\mathbf{z}_t = q(\mathbf{z}_0, \epsilon, t) = \alpha_t \mathbf{z}_0 + \sigma_t \epsilon, \quad \forall t \in [0, T], \tag{4}$$

where $\alpha_t$ and $\sigma_t$ define the signal-to-noise ratio (SNR) of the stochastic interpolant $\mathbf{z}_t$. Following previous work (Chen et al., 2023; Liang et al., 2024), we opt for coefficients $(\alpha_t, \sigma_t)$ that satify $\alpha_t + \sigma_t^2 = 1$, resulting in a variance-preserving process (Karras et al., 2022). Under the continuous time limit, the forward process in Eq. 4 is equal to the stochastic differential equation (SDE): $d\mathbf{z} = \mathbf{f}(\mathbf{z}, t)dt + g(t)d\mathbf{w}_t$, where $f(t) = \frac{\text{d} \log \alpha_t}{\text{d}t}$ is a drift coefficient, $g^2(t) = \frac{\text{d}\sigma_t^2}{\text{d}t} - 2\frac{\text{d} \log \alpha_t}{\text{d}t}\sigma_t^2$ is the diffusion coefficient.

**Backward Process**. Inversely, the backward diffusion process is intended to reverse the noising process and generate samples. According to Anderson's theorem (Anderson, 1982), the forward SDE introduced earlier satisfies a reverse-time diffusion equation, which can be reformulated using the Fokker-Planck equations (Song et al., 2020) to have a deterministic counterpart with equivalent marginal probability densities, known as the *probability flow ODE* (PF-ODE): $d\mathbf{z} = \left[ \mathbf{f}(\mathbf{z}, t) - \frac{1}{2}g(t)^2 \nabla_\mathbf{z} \log p_t(\mathbf{z}) \right] dt$. As demonstrated in (Hyvärinen & Dayan, 2005; Song et al., 2020), this marginal transport map can be learned through maximum likelihood estimation of the perturbation kernel of diffused data samples $\nabla_\mathbf{z} \log p_t(\mathbf{z}|\mathbf{z}_0)$ in a simulation-free manner. This allows us to estimate $\hat{\epsilon}(\mathbf{z}_t, t)/\sigma_t \approx \nabla_\mathbf{z} \log p_t(\mathbf{z}|\mathbf{z}_0)$, usually parameterized by a time-conditioned network.

**Exact Solution of PF-ODEs**. Given an initial value $z_s$ at time $s > 0$, the solution $z_t$ at time $t \in [0, s]$ of diffusion ODEs in Eq. 3.2 is expressed as a *semi-linear* ODE:

$$z_t = \underbrace{e^{\int_s^t f(r)dr} z_s}_{\text{linear part}} + \int_s^t \underbrace{\left( e^{\int_r^t f(r)dr} \frac{g^2(\tau)}{2\sigma_\tau} \epsilon_\theta(z_\tau, \tau) \right)}_{\text{non-linear part}} d\tau, \tag{5}$$

It decouples the linear part and the nonlinear part. Unlike the black-box ODE solvers (Karras et al., 2022), the linear part is exactly computed (*i.e.* $f(t) = \frac{\text{d} \log \alpha_t}{\text{d}t}, e^{\int_s^t f(r)dr} = log\frac{\alpha_t}{\alpha_s}$), which eliminates the approximation error of the linear term. However, the integral of the nonlinear component remains complex, as it couples the coefficients related to the noise schedule (i.e., $f(\tau), g(\tau), \sigma_\tau$) with the intricate neural network $\epsilon_\theta$, making it difficult to approximate.

**Second-order ODE-Solver using exact solution.** The commonly used solver DDIM could be obtained by deriving the first-order Taylor expansion formulae on Eq. 5's non-linear part. Numerous

numerical solvers (Lu et al., 2022; Zhao et al., 2023) exist for approximating the nonlinear component, their performance varies significantly with large step sizes (fewer sampling steps). This motivated our development of a possible efficient discretization scheme balancing fidelity and speed. Empirically (See Appendix C for details), we employ Diffusion Exponential Integrator Sampler (DEIS) (Zhang & Chen, 2022) as the second-order approximation of Eq. 5's non-linear part, with respect to $t$, which minimizes error and achieves superior quality with 10 denoising steps:

$$z_{t-1} = \frac{\alpha_t}{\alpha_{t-1}} z_t + \frac{\rho_t}{\log \rho_{t-1} - \log \rho_{t-2}} \left[ (\log \rho_t - \log \rho_{t-1}) \epsilon_t - (\log \rho_t - \log \rho_{t-2}) \epsilon_{t+1} \right], \quad (6)$$

where $\rho_t = \frac{\sigma_t}{\alpha_t}$, and $\epsilon_t$ represent the output of denoiser at timestep $t$.

**Neural Implicit Collision Guidance.** Compared to single-agent motion generation, mesh collision artifacts between humans cause unique challenges for multi-person motion generation. Since the interaction representation includes no explicit mesh information, it is difficult to constrain the inter-person collision in the training process. Current methods (Jiang et al., 2025b; Ota et al., 2025) alleviate this by adapting post-hoc optimization methods based on differentiable objectives, most of which rely on the occupancy-based or explicit mesh-based detection.

However, such optimization is executed between different sampling steps, which inevitably incurs additional computational cost and thus increases the per-step sampling latency of ILD if adapted. To address this, we introduce an efficient neural collision guidance based on VolumetricSMPL (Mihajlovic et al., 2025), which extends the SMPL (Loper et al., 2015) by representing human shape $\beta$ as a Signed Distance Field (SDF). By leveraging the Neural Blend Weights (Mihajlovic et al., 2023) generator, it significantly reduces the computational costs compared to the previous neural implicit detection (*i.e.* COAP (Mihajlovic et al., 2022)) and mesh-based detection (Jiang et al., 2020).

While VolumetricSMPL (Mihajlovic et al., 2025) provides an efficient SDF representation, it is not explicitly designed for human-human collision. We therefore define a targeted interpenetration loss by querying the SDFs of the two bodies' surface A and B: $\mathcal{L}_{ip} = \sum_{v_a \in V_A} -\tilde{d}_B(v_a | \beta_b, \theta_b)$, as the sum of penetration depths over all colliding vertices. During the denoising step $z_{t-1} = \text{DEIS}(z_t)$, we decode $z_{t-1}$ to poses and detect interpenetrations implicitly. (See Appendix D for Algorithm)

## 3.3 FLASH INTERACTION DIFFUSION DISTILLATION

While optimizing the latent diffusion training and improving the efficiency of the ODE sampler, ILD requires 10 steps for satisfying interaction generation due to fidelity degeneration (see Tab. 2d). To achieve real-time inference (*i.e.*, 30 FPS), we introduce Flash Interaction Latent Diffusion (FILD), which consists of Interaction Consistency Models and Interaction Distribution Matching. As shown in **Stage 3** of Fig. 2, FILD trains a student model based on the constraints from the frozen ILD teacher denoiser, enabling single-step inference with preserved interaction generation quality.

**Interaction Consistency Model.** Similar to previous works (Dai et al., 2024; Jiang et al., 2025a), FILD employs a consistency model (CM) (Song et al., 2023; Luo et al., 2023) as the backbone scheduler to distill the student network, leveraging the knowledge from the teacher denoiser. The consistency function is defined as $f : (\mathbf{z}_t, t) \mapsto \mathbf{z}_\epsilon$, where $0 < \epsilon \ll T$ (such as 0.002), with $\hat{\mathbf{z}}_\epsilon \sim p_{\text{data}}(\mathbf{z})$ and $z_t$ is the noisy latent vector at timestep t. The self-consistency of the function is expressed as: $f(z_t, t) = f(z_{t'}, t'), \quad \forall t, t' \in [\epsilon, T]$. The student consistency model, $f_\Theta(\cdot, \cdot)$, aims to learn this property by training: $f_\Theta(z_t, t) = c_{\text{skip}}(t) z_t + c_{\text{out}}(t) F_\Theta(z_t, t)$, where $c_{\text{skip}}(t)$ and $c_{\text{out}}(t)$ are differentiable functions with $c_{\text{skip}}(\epsilon) = 1$ and $c_{\text{out}}(\epsilon) = 0$. We leverage a pretrained teacher denoiser $\hat{\Phi}$ to intially parameterize $F_\Theta(z_t, t) = \Phi(x, t)$ for self-consistency learning. The consistency loss for distillation learning is defined as:

$$\mathcal{L}_{\text{distill}}(\Theta) = \mathbb{E}[d(f_\Theta(z_{t_n}, t_n), f_{\hat{\Phi}}(\hat{z}_{t_n}^{\hat{\Phi}}, t_n))], \quad (7)$$

where $d(\cdot, \cdot)$ is a chosen metric function for measuring the distance between two samples. $f_\Theta(\cdot, \cdot)$ and $f_{\hat{\Phi}}(\cdot, \cdot)$ are referred to as "online network" and "target network" respectively (Song et al., 2023).

Please note that it is non-trivial to conduct distillation learning for interaction generation by following (Dai et al., 2024), which causes artifacts and instability. Thus, we introduce two key improvements: 1) We skip the classifier-free guidance (CFG) distillation (Meng et al., 2023) for the student network, which hampers one-step generation performance (see Sec. 2b). 2) We redefine $\hat{z}^{\hat{\Phi}}$ as a full chain diffusion reverse process of the teacher denoiser instead of the one-step estimation used in MotionLCM (Dai et al., 2024):

$$f_{\hat{\Phi}}(\hat{z}_{t_n}^{\hat{\Phi}}, t_n) \leftarrow \mathbf{z}_{t_n} + \sum_{i=0}^{n} (t_i - t_{i-k}) \hat{\Phi}(\mathbf{z}_{t_i}, t_i, c; \theta), \quad (8)$$

where $\Phi$ acts as the pretrained ILD denoiser with the DDIM solver. With these improvements, the student network generates interaction sequences more efficiently in fewer steps, closely matching the output of the pretrained ILD teacher denoiser with multi-step classifier-free guidance.

**Interaction Distribution Matching.** We also introduce Distribution Matching Distillation (DMD) (Wang et al., 2023; Yin et al., 2024) training to improve the capability of the student network. The DMD constraint is designed to ensure the generated samples closely mirror the interaction distribution learned by the teacher denoiser, minimizing the KL divergence between the student distribution $p_\theta^{stu}$ and the teacher distribution $p_\theta^{tea}$ as: $\mathcal{L}_{\text{DMD}} = D_{\text{KL}}(p_\theta^{\text{stu}} \| p_\phi^{\text{tea}})$. According to Wang et al. (2023), the gradient of the KL divergence to the student consistency model $f_\Theta(z_t, t)$ is:

$$\nabla_\Theta \mathcal{L}_{\text{DMD}} = \mathbb{E}\left[\left(s^{\text{stu}}(y) - s^{\text{tea}}(y)\right) \nabla_{f_\Theta(z_t, t)}\right], \qquad (9)$$

where $s^{\text{tea}}$ and $s^{\text{stu}}$ are the score functions of teacher denoiser $\Phi$ and student denoiser $\Phi$, and $y = f_\Theta(z_t, t)$ is the consistency model output in Eq. 3.3. However, it remains challenging to compute this gradient (Yin et al., 2024) since the scores diverge for samples with low probability, specifically, the teacher distribution is highly likely to vanish for fake samples. Thus, the one-step student prediction $y$ is re-noised using a uniformly sampled timestep $t' \sim \mathcal{U}([0, 1])$ and the teacher noise schedule. The new noisy sample is passed through the frozen teacher denoiser to get the score function for the teacher distribution $s^{\text{tea}}(f_\theta(z_{t'}, t')) = -(\epsilon_\phi^{\text{tea}}(z_{t'}, t')/\sigma(t'))$, according to (Karras et al., 2022). Rather than another dedicated diffusion model (Chadebec et al., 2024), we utilise the student model for the score function of the student distribution $s^{\text{stu}}(f_\theta(z_{t'}, t')) = -(\epsilon_\phi^{\text{stu}}(z_{t'}, t')/\sigma(t'))$.

Taken together, the proposed FILD is trained to minimize a weighted combination of the distillation loss and the distribution matching losses: $\mathcal{L}_{\mathcal{FILD}} = \mathcal{L}_{\text{distil}} + \lambda_{\text{DMD}} \mathcal{L}_{\text{DMD}}$.

# 4 EXPERIMENTS

## 4.1 DATASETS

We utilize InterHuman (Liang et al., 2024) and Inter-X (Xu et al., 2024) datasets for the evaluation of text-to-interaction generation performance, which contain 7,779 and 11,388 interaction sequences, respectively. Each interaction sequence is annotated with 3 textual descriptions.

InterHuman follows the SMPL (Loper et al., 2015) skeleton representation with 22 joints, including the root joint. Each interaction sequence in a certain frame $i_{th}$ could be represented by $x^i = \left[\mathbf{j}_g^p, \mathbf{j}_g^v, \mathbf{j}^r, \mathbf{c}^f\right]$, which is the collection of joint positions $\mathbf{j}_g^p \in \mathbb{R}^{2 \times 22 \times 3}$, joint velocities $\mathbf{j}_g^v \in \mathbb{R}^{2 \times 22 \times 3}$, 6D representation rotations $\mathbf{j}^r \in \mathbb{R}^{2 \times 21 \times 6}$, and binary foot-ground contact features $\mathbf{c}^f \in \mathbb{R}^{2 \times 4}$, resulting in a total input dimension of 524.

Inter-X follows the SMPL-X (Pavlakos et al., 2019) skeleton representation, comprising 54 body and hand joints, accompanied by root orientation and translation. Each interaction sequence could be represented by $x^i = \left[\mathbf{r}_g^p, \mathbf{j}^r\right]$, and which is the collection of root joint positions $\mathbf{r}_g^p \in \mathbb{R}^{2 \times 3}$, Euler Angle representation rotations $\mathbf{j}^r \in \mathbb{R}^{2 \times 55 \times 3}$, resulting in a total input dimension of 336.

## 4.2 EVALUATION METRICS

We employ the evaluation metrics following previous studies (Liang et al., 2024; Guo et al., 2022). Fidelity is assessed using Frechet Inception Distance (FID), R-precision, and Multimodal Distance (MM Dist), and diversity is evaluated with Diversity and Multimodality scores. We evaluate the collision metric via winding number (Mihajlovic et al., 2022). (See Appendix D for implementation and training details, Appendix E for metric definition, and Appendix F for hyperparameter ablation)

## 4.3 RESULTS

### 4.3.1 QUANTITATIVE RESULTS

In Table 1, we present the overall evaluation with respect to fidelity and diversity metrics. For the InterHuman dataset (Liang et al., 2024), the proposed ILD has achieved the best 'FID' and competitive 'R Precision' results. Notably, ILD presents a very close 'Multimodal Dist' with SoTA models. For the Inter-X dataset (Liang et al., 2024), the proposed ILD excels across all fidelity metrics except for the 'MModality' metrics. FILD keeps the second-best performance in terms of 'R Precision' and 'FILD'. Overall, the diversity metric 'Multimodality' shows room for improvement. We hypothesize that freezing the IA-VAE may reduce diversity in generations for unseen text descriptions, but this can be a favorable trade-off in applications that prioritize precise and realistic motion synthesis.

### 4.3.2 QUALITATIVE RESULTS

Fig. 3 demonstrates ILD's ability to generate more realistic human interactions compared to Inter-Mask (Javed et al., 2024). In the "taekwondo" scenario, ILD correctly synthesizes dynamic attack-

Table 1: Quantitative evaluation on the InterHuman and InterX test sets. $\pm$ indicates a 95% confidence interval and $\rightarrow$ means the closer to ground truth the better. **Bold face** indicates the best result, while underline refers to the second best. '†' refers to TIMotion based on the Transformer.

| Dataset | Method | R Precision↑ | | | FID↓ | MM Dist↓ | Diversity→ | MModality↑ |
|---|---|---|---|---|---|---|---|---|
| | | Top 1 | Top 2 | Top 3 | | | | |
| | Ground Truth | $0.452^{\pm0.008}$ | $0.610^{\pm0.009}$ | $0.701^{\pm0.008}$ | $0.273^{\pm0.007}$ | $3.755^{\pm0.008}$ | $7.948^{\pm0.064}$ | - |
| Inter Human | TEMOS (Petrovich et al., 2022) | $0.224^{\pm0.010}$ | $0.316^{\pm0.013}$ | $0.450^{\pm0.018}$ | $17.375^{\pm0.043}$ | $6.342^{\pm0.015}$ | $6.939^{\pm0.071}$ | $0.535^{\pm0.014}$ |
| | T2M (Guo et al., 2022) | $0.238^{\pm0.012}$ | $0.325^{\pm0.010}$ | $0.464^{\pm0.014}$ | $13.769^{\pm0.072}$ | $5.731^{\pm0.013}$ | $7.046^{\pm0.022}$ | $1.387^{\pm0.076}$ |
| | MDM (Tevet et al., 2022b) | $0.153^{\pm0.012}$ | $0.260^{\pm0.009}$ | $0.339^{\pm0.012}$ | $9.167^{\pm0.056}$ | $7.125^{\pm0.018}$ | $7.602^{\pm0.045}$ | $\mathbf{2.350^{\pm0.080}}$ |
| | ComMDM (Shafir et al., 2023) | $0.223^{\pm0.009}$ | $0.334^{\pm0.008}$ | $0.466^{\pm0.010}$ | $7.069^{\pm0.054}$ | $6.212^{\pm0.021}$ | $7.244^{\pm0.038}$ | $1.822^{\pm0.052}$ |
| | InterGen (Liang et al., 2024) | $0.371^{\pm0.010}$ | $0.515^{\pm0.012}$ | $0.624^{\pm0.010}$ | $5.918^{\pm0.079}$ | $5.108^{\pm0.014}$ | $7.387^{\pm0.029}$ | $\underline{2.141^{\pm0.063}}$ |
| | in2IN (Ponce et al., 2024) | $0.425^{\pm0.008}$ | $0.576^{\pm0.008}$ | $0.662^{\pm0.009}$ | $5.535^{\pm0.120}$ | $3.803^{\pm0.002}$ | $\underline{7.953^{\pm0.047}}$ | $1.215^{\pm0.023}$ |
| | TIMotion† (Wang et al., 2025) | $\underline{0.491^{\pm0.005}}$ | $\mathbf{0.648^{\pm0.004}}$ | $\mathbf{0.724^{\pm0.004}}$ | $5.433^{\pm0.080}$ | $\mathbf{3.775^{\pm0.001}}$ | $8.032^{\pm0.030}$ | $0.952^{\pm0.032}$ |
| | InterMask (Ponce et al., 2024) | $0.449^{\pm0.004}$ | $0.599^{\pm0.005}$ | $0.683^{\pm0.005}$ | $5.154^{\pm0.061}$ | $3.790^{\pm0.002}$ | $\mathbf{7.944^{\pm0.033}}$ | $1.737^{\pm0.020}$ |
| | ILD | $\mathbf{0.495^{\pm0.005}}$ | $0.630^{\pm0.005}$ | $\underline{0.709^{\pm0.004}}$ | $\mathbf{4.869^{\pm0.073}}$ | $\underline{3.777^{\pm0.001}}$ | $7.976^{\pm0.027}$ | $0.881^{\pm0.022}$ |
| | FILD | $0.484^{\pm0.005}$ | $\underline{0.636^{\pm0.004}}$ | $0.701^{\pm0.003}$ | $\underline{4.980^{\pm0.041}}$ | $3.780^{\pm0.001}$ | $8.024^{\pm0.032}$ | $1.124^{\pm0.020}$ |
| | Ground Truth | $0.429^{\pm0.004}$ | $0.626^{\pm0.003}$ | $0.736^{\pm0.003}$ | $0.002^{\pm0.002}$ | $3.536^{\pm0.013}$ | $9.734^{\pm0.078}$ | - |
| InterX | TEMOS (Petrovich et al., 2022) | $0.092^{\pm0.003}$ | $0.171^{\pm0.003}$ | $0.238^{\pm0.002}$ | $29.258^{\pm0.069}$ | $6.867^{\pm0.013}$ | $4.738^{\pm0.078}$ | $0.672^{\pm0.041}$ |
| | T2M (Guo et al., 2022) | $0.184^{\pm0.010}$ | $0.298^{\pm0.006}$ | $0.396^{\pm0.005}$ | $5.481^{\pm0.382}$ | $9.576^{\pm0.006}$ | $5.771^{\pm0.151}$ | $2.761^{\pm0.042}$ |
| | MDM (Tevet et al., 2022b) | $0.203^{\pm0.009}$ | $0.329^{\pm0.007}$ | $0.426^{\pm0.005}$ | $23.701^{\pm0.057}$ | $9.548^{\pm0.014}$ | $5.856^{\pm0.077}$ | $\underline{3.490^{\pm0.061}}$ |
| | ComMDM (Shafir et al., 2023) | $0.090^{\pm0.002}$ | $0.165^{\pm0.004}$ | $0.236^{\pm0.004}$ | $29.266^{\pm0.067}$ | $6.870^{\pm0.017}$ | $4.734^{\pm0.067}$ | $0.771^{\pm0.053}$ |
| | InterGen (Liang et al., 2024) | $0.207^{\pm0.004}$ | $0.335^{\pm0.005}$ | $0.429^{\pm0.005}$ | $5.207^{\pm0.216}$ | $9.580^{\pm0.011}$ | $7.788^{\pm0.208}$ | $\mathbf{3.686^{\pm0.052}}$ |
| | TIMotion† (Wang et al., 2025) | $\underline{0.412^{\pm0.004}}$ | $0.601^{\pm0.004}$ | $0.714^{\pm0.003}$ | $0.385^{\pm0.218}$ | $3.706^{\pm0.015}$ | $\underline{9.191^{\pm0.092}}$ | $2.437^{\pm0.069}$ |
| | InterMask (Ponce et al., 2024) | $0.403^{\pm0.005}$ | $0.595^{\pm0.004}$ | $0.705^{\pm0.005}$ | $0.399^{\pm0.013}$ | $3.705^{\pm0.017}$ | $9.046^{\pm0.073}$ | $2.261^{\pm0.081}$ |
| | ILD | $\mathbf{0.441^{\pm0.005}}$ | $\mathbf{0.621^{\pm0.004}}$ | $\mathbf{0.733^{\pm0.004}}$ | $\mathbf{0.297^{\pm0.012}}$ | $\mathbf{3.568^{\pm0.028}}$ | $\mathbf{9.253^{\pm0.067}}$ | $1.931^{\pm0.024}$ |
| | FILD | $0.424^{\pm0.005}$ | $\underline{0.603^{\pm0.005}}$ | $\underline{0.728^{\pm0.004}}$ | $\underline{0.305^{\pm0.010}}$ | $\underline{3.667^{\pm0.012}}$ | $8.944^{\pm0.072}$ | $2.168^{\pm0.044}$ |

**ILD**        **InterMask**

*The two guys are practicing taekwondo, and one person is executing offensive moves while the other person is evading the attacks."*

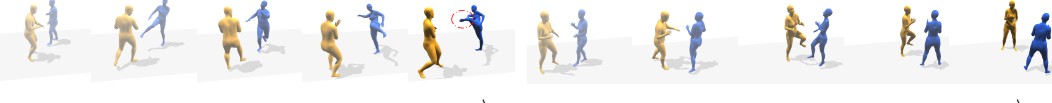

*These two people face each other. One person approaches and hands over an item to the other.*

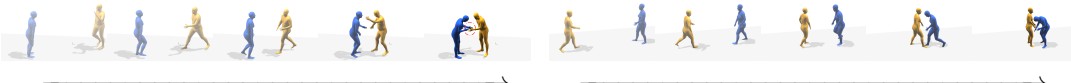

*The other person bends down to help the first one up and puts both hands on their chest to support them walking to the side.*

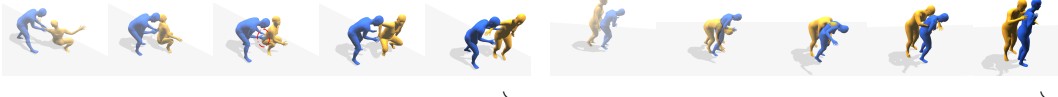

Figure 3: Qualitative comparison (zoom in to see it better) between InterMask (Javed et al., 2024) and ILD, highlighting ILD's superior interaction quality and text adherence. The visualization is based on aitviewer software (Kaufmann et al., 2022). The deeper colors indicate the later in time.

and-evade motions with combative contact, while InterMask ignores its semantics with only generic motions. For the "hand over" prompt, ILD models the explicit hand-to-hand exchange between two people. In contrast, InterMask's result is ambiguous, in which only one person shows the transfer gesture. In the complex "helping up" interaction, ILD generates a physically plausible supportive lift. InterMask's output suffers from severe interpenetration, and the characters bend in parallel, failing to form a supportive connection.

### 4.4 ABLATION STUDIES

*ILD Component Contribution Analysis.* Table 2a shows that removing the designed loss functions degrades ILD's performance, particularly the interactive loss. We also compare it with the InterLDM (Li et al., 2024a), which only adapts the geometric loss. The results confirm the critical importance of modeling relationships between individuals when generating interaction sequences.

*FILD Component Contribution Analysis.* We compare with MotionLCM (Dai et al., 2024), which also utilises the consistency models for real-time individual motion generation. Table 2b shows that omitting the conditioning training of the classifier-free guidance scale can help stabilise distillation training. The key improvement comes from a full chain diffusion backwards rather than a fixed

Table 2: Ablation results on the InterHuman dataset.

| Methods | FID ↓ | R Top 3 ↑ |
|---|---|---|
| InterLDM | 5.619 | 0.638 |
| ILD | 4.869 | 0.709 |
| w/o Int. Loss | 6.145 | 0.671 |
| w/o Geo. Loss | 5.189 | 0.680 |
| w/o KL. Loss | 5.454 | 0.677 |

(a) The ablation study for interaction-aware loss.

| Methods | FID ↓ | R Top 3 ↑ |
|---|---|---|
| MotionLCM | 10.262 | 0.591 |
| FILD | 4.980 | 0.701 |
| w/o CFG Scale | 6.301 | 0.666 |
| w/o Full Chain | 7.209 | 0.649 |
| w/o DMD. Loss | 5.724 | 0.654 |

(b) The ablation study to verify key components of the proposed FILD.

| Methods | Collision ↓ | AITS ↓ |
|---|---|---|
| InterGen | 0.989 | 2.892 |
| ILD | 0.913 | 0.097 |
| w. SDF | 0.557 | 1.025 |
| w. COAP | 0.297 | 4.793 |
| w. VM (ours) | 0.264 | 0.564 |

(c) Physical performance for different collision detection.

| Solver | Steps | Method | FID↓ | AITS↓ |
|---|---|---|---|---|
| DDIM | 10 | in2IN | 5.927 | 0.608 |
| DEIS | 10 | in2IN | 5.712 | 0.614 |
| DDIM | 10 | ILD | 5.263 | 0.093 |
| DPM | 10 | ILD | 5.037 | 0.097 |
| DEIS | 10 | ILD | 4.869 | 0.097 |
| DEIS | 1 | in2IN | 19.323 | 0.126 |
| DEIS | 1 | ILD | 12.292 | 0.034 |
| CM | 1 | FILD | 4.980 | 0.032 |

(d) Influence of the ODE solver and sampling steps on computational efficiency.

| Dimension | Alignment | rFID↓ | gFID↓ | AITS↓ |
|---|---|---|---|---|
| $1 \times 512$ | ✗ | 0.776 | 4.981 | 0.089 |
| $2 \times 512$ | ✗ | 0.412 | 5.237 | 0.093 |
| $4 \times 512$ | ✗ | 0.185 | 5.214 | 0.095 |
| $8 \times 512$ | ✗ | 0.134 | 5.931 | 0.119 |
| $1 \times 512$ | ✓ | 0.998 | 4.967 | 0.089 |
| $2 \times 512$ | ✓ | 0.493 | 5.097 | 0.094 |
| $4 \times 512$ | ✓ | 0.212 | 4.869 | 0.097 |
| $8 \times 512$ | ✓ | 0.159 | 5.511 | 0.125 |

(e) Influence of the ILD latent dimension and alignment on VAE (rFID) and diffusion (gFID) fidelity.

interval step (such as 10 in Dai et al. (2024)) for training consistency models. Additionally, DMD loss could further enhance it through the adversarial learning scheme.

*Physical Plausibility and Efficiency.* As shown in Table 2c and Fig. 4, compared to the InterGen (Liang et al., 2024) without any post-hoc optimization, the collision guidance (Karunratanakul et al., 2023) substantially reduces interpenetration. Our implicit model based on VolumetricSMPL (Mihajlovic et al., 2025) generates more physically plausible motions than mesh-based SDF (Jiang et al., 2020) methods. Crucially, improved physical accuracy does not come at a significant cost. Our method's optimization is $2\times$ faster than SDF and $9\times$ faster than COAP (Mihajlovic et al., 2022), respectively, using the same optimization iterations and hardware (NVIDIA A100).

*Sampling Method Efficiency Comparison.* As shown in Table 2d, we compare the proposed methods with the SoTA diffusion model in2IN (Ponce et al., 2024), ILD shows minimal degradation during few-step sampling due to its interaction-aware latent space. Moreover, DEIS (Zhang & Chen, 2022) solver consistently outperforms DDIM in few-step settings for both motion-space and latent-space diffusion models (see Appendix H for computational resource comparison).

*Dimension size and Tokenizer Alignment Analysis* Table 2e details our analysis of the latent space dimensionality. We observe that the higher dimensions improve reconstruction but reduce generation quality. Our proposed tokenizer alignment loss effectively mitigates this issue, enhancing generative performance in high-dimensional settings. Notably, due to in-context learning for the text condition, the increased dimension causes minor computational delays for diffusion generation, where only negligible overhead cost stems from the linear projection required to align the dimensions of the RQ-VAE (Guo et al., 2024) and the IA-VAE.

## 5 CONCLUSION

In this work, we present Interaction Latent Diffusion (ILD) and its real-time variant, Flash ILD (FILD), to generate complex, multi-person interactions from text. Our core contribution is an interaction-aware, multi-token latent space embedded in ILD, unlocking the full capacity of the diffusion model. By constraining with inter-person relationships and aligning with a pretrained motion tokenizer, we enhance its expressive capacity without destabilizing the diffusion process. To achieve efficient and physically plausible synthesis, we couple a high-order ODE solver with a lightweight neural collision guidance, enabling high-fidelity generation in a few steps. Building on this, FILD distills the learned ILD into a one-step generator via a tailored consistency and distribution matching pipeline. Extensive evaluations on the InterHuman and Inter-X datasets demonstrate that our work offers a robust framework for efficient dyadic interaction generation, achieving a balance between quality and speed. Moving forward, we aim to integrate more sophisticated physical constraints directly into training.

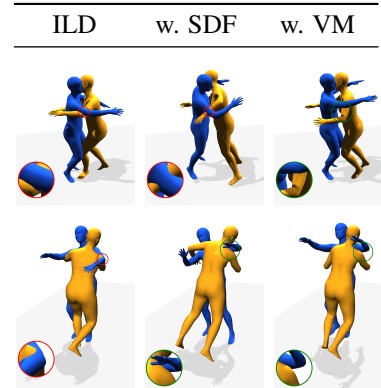

Figure 4: Optimization 'Hugging' with different collision detection, including two views, where green circles indicate correct collision optimization, while red circles denote incorrect optimization.

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

# Appendix

This appendix provides a detailed explanation of our loss functions (Sec. A), more motion tokenizer alignment qualitative results (Sec. B), the empirical study for different ODE solvers (Sec. C), the algorithms for ILD sampling (Sec. D), implementation details (Sec. E), details of metric definitions (Sec. F), ablation studys (Sec. G) on classifier-free guidance hyperparameters, batch size and loss weights and a computational resource comparison (Sec. H). We also provide a statement for LLM usage in Sec. I.

**Video.** In the supplementary video, we show 1) more comparisons of text-to-interaction generation, 2) more ablation study cases, and 3) additional samples of failure cases. We suggest the reader watch this video for dynamic motion results.

## A INTERACTIVE AND GEOMETRIC LOSS FUNCTION

In a typical VAE training process, motion reconstruction $x^{1:N}$ is constrained by the Mean Squared Error (MSE) and Kullback-Leibler (KL) losses. To enhance the physical plausibility within involved individuals and preserve the original interaction relationships between individuals, we further adapted the geometric loss and interactive loss.

Firstly, motion generation is generally regularized using geometric loss (Tevet et al., 2022b), which enforces physical plausibility and prevents artifacts from generating smooth and natural motions. In this work, we applied the Bone length (BL) loss and foot contacting loss as follows:

$$\mathcal{L}_{BL} = \|B\left(\hat{x}_a\right) - B\left(x_a\right)\|_2^2 + \|B\left(\hat{x}_b\right) - B\left(x_b\right)\|_2^2, \tag{10}$$

$$\mathcal{L}_{\text{foot}} = \frac{1}{N-1} \sum_{i=1}^{N-1} \left\| \left( FK\left(\hat{x}_{foot}^{i+1}\right) - FK\left(\hat{x}_{foot}^i\right) \right) \cdot f_i \right\|_2^2 \tag{11}$$

$$\mathcal{L}_{\text{geometric}} = \lambda_{\text{BL}}\,\mathcal{L}_{\text{BL}} + \lambda_{\text{FC}}\,\mathcal{L}_{\text{foot}} \tag{12}$$

where $B$ represents the bone lengths in a predefined human body kinematic tree derived from the global joint positions, and $FK$ denotes the forward kinematic function converting joint rotations into joint positions. Bone length loss $\mathcal{L}_{\text{BL}}$ constrains the global joint positions of each person to satisfy skeleton consistency, which implicitly encodes the human body's kinematic structure. $f_i \in \{0,1\}^J$ is the binary foot contact mask for each frame i, indicating whether they touch the ground; it mitigates the foot-sliding effect by nullifying velocities when touching the ground.

Secondly, to handle the complexity of spatial relations in multi-person interactions, we further introduce interactive losses, comprising masked joint distance map (DM) loss and relative orientation (RO) loss (Liang et al., 2024) as follows:

$$\mathcal{L}_{DM} = \left\| (M(\hat{\mathbf{x}}_a, \hat{\mathbf{x}}_b) - M(\mathbf{x}_a, \mathbf{x}_b)) \odot I(M_{xz}(\mathbf{x}_a, \mathbf{x}_b) < \bar{M}) \right\|_2^2 \tag{13}$$

$$\mathcal{L}_{RO} = \|O_y\left(IK\left(\hat{\mathbf{x}}_a\right), IK\left(\hat{\mathbf{x}}_b\right)\right) - O_y\left(IK\left(\mathbf{x}_a\right), IK\left(\mathbf{x}_b\right)\right)\|_2^2 \tag{14}$$

$$\mathcal{L}_{\text{interactive}} = \lambda_{\text{DM}}\,\mathcal{L}_{\text{DM}} + \lambda_{\text{RO}}\,\mathcal{L}_{\text{RO}} \tag{15}$$

Regarding the DM loss, we first measure the $N_j \times N_j$ joint distance map between two generated individual motions and then match it with the ground truth. Besides, we activate this loss only when the horizontal distance between the two generated individual motions is small enough, which could stabilize the training process. As shown in Equation (13), $I$ is the indicator function that masks the loss by applying a 2D distance threshold on the $XZ$-plane, $M_{xz}$ represents the distance map projected onto the $XZ$-plane, $\overline{M}$ is the distance threshold, and $O$ indicates the Hadamard product. The RO loss estimates the relative root orientation of two people and aligns it with the ground truth. As shown in Equation (14), $IK$ represents the inverse kinematics process, which outputs the joint rotations, and $O_y$ indicates the 2D root relative orientation between the two people around the $Y$-axis obtained from rotations.

Specifically, in the DM loss, 'M' is the distance map in world coordinates. Instability stems from large distance variations, and we set a low threshold $\overline{M}$ to alleviate it and enhance close-interaction generation. In the RO loss, the root rotation is not provided in the InterHuman representation ($j^r \in \mathbb{R}^{2 \times 21 \times 6}$). To calculate it, we compute the normalized hip vector $\vec{a} = \frac{\vec{j}_{rh} - \vec{j}_{lh}}{\|\vec{j}_{rh} - \vec{j}_{lh}\|}$, where $\vec{j}_{rh}, \vec{j}_{lh} \in \mathbb{R}^3$ are right and left hip positions. With unit y axis vector $\vec{y} = [0, 1, 0]$, the forward vector is derived as $\vec{f} = \frac{\vec{y} \times \vec{a}}{\|\vec{y} \times \vec{a}\|}$. The orientation difference between predicted and ground truth forward vectors is then used to compute the RO loss in the XZ plane, the process is differentiable.

## B  MOTION TOKENIZER

Table 3: The influence of motion alignment loss on reconstruction and generation performance. The latent dimension here is set as $4 \times 512$.

| Tokenizer | Reconstruction Performance | | | Generation Performance↓ | | |
|---|---|---|---|---|---|---|
| | rFID↓ | MPJPE↓ | MPJVE ↓ | gFID↓ | gRTop3↑ | AITS↓ |
| ILD w/o. MA loss | 0.185 | 10.05 | 8.47 | 5.214 | 0.673 | 0.095 |
| ILD w. MA loss (*VQ-VAE*) | 0.306 | 11.43 | 8.98 | 5.687 | 0.630 | 0.095 |
| ILD w. MA loss (*RVQ-VAE*) | 0.212 | 9.79 | 8.36 | 4.869 | 0.709 | 0.097 |
| Interpolation | 0.297 | 10.56 | 8.61 | 5.309 | 0.644 | 0.095 |
| Pooling | 0.341 | 10.13 | 8.54 | 6.313 | 0.621 | 0.096 |
| Linear layer transformation | 0.212 | 9.79 | 8.36 | 4.869 | 0.709 | 0.097 |

## C  ODE SOLVER EMPIRICAL STUDY

Our empirical analysis validates three key insights: 1) *Higher-order solvers reduce errors in few-step settings.* Fig. 5 shows that the second-order samplers like DEIS (Zhang & Chen, 2022) outperform, especially in the few-step settings, while DDIM maintains its potential in many-step sampling. 2) *Interaction-aware latent space enhances exact PF-ODE solver advantages.* Interestingly, for the ILD trained without interaction-aware space on Fig. 6, DDIM shows a higher sampling error than all other ODE solvers, even including HeuD solver (Karras et al., 2022) with its non-analytical solution. 3) *Higher-order solvers improve diffusion distillation.* Based on Fig. 7, we also find that High-order ODE solvers boost FILD performance, with the ILD teacher using UniPC (Zhao et al., 2023) solver providing the most accurate cases for the student to mimic.

## D  ILD SAMPLING ALGORITHM

---

**Algorithm 1:** ILD Sampling

---

**Input:** Prompt $\mathcal{T}$, steps $N$, params $K$, body shape $\theta$, $\eta$
**Output:** Motion sequence $x_I$

---

1   $c \leftarrow \mathcal{E}_{\text{text}}(\mathcal{T})$, $z_N \sim \mathcal{N}(0, \mathbf{I})$;
2   $\epsilon_{t_{N+1}} \leftarrow \epsilon_\theta(z_N, t_N, c)$;
3   **for** $i = N$ **to** 1 **do**
4     $z_{i-1} \leftarrow \text{DEIS}(z_i, \epsilon_{t_i}, \epsilon_{t_{i+1}})$;
5     $x_{i-1} \leftarrow \mathcal{D}_{\text{IA}}(z_{i-1})$; // Decode
6     **for** $k = 1$ **to** $K$ **do**
7       $\beta \leftarrow x_{i-1}$; $\nabla \leftarrow \nabla_{z_{i-1}} \text{VM}(\beta, \theta)$;
8       $z_{i-1} \leftarrow z_{i-1} - \eta \cdot \nabla / \|\nabla\|$;
9     **end**
10 **end**
11 **return** $\mathcal{D}_{IA}(z_0)$; // Decode

---

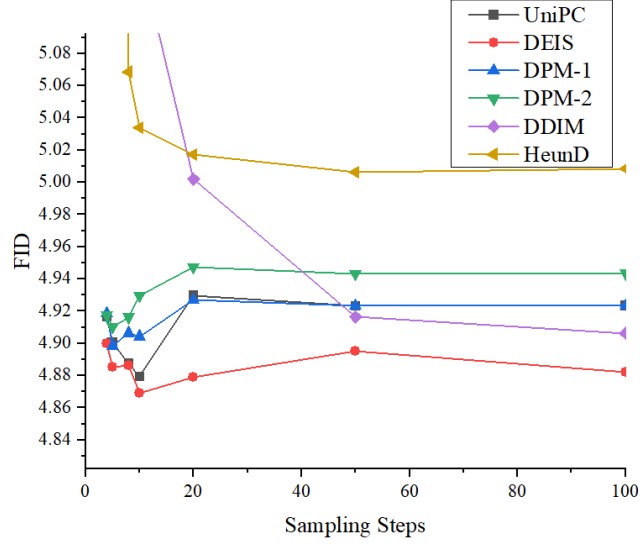

Figure 5: ILD

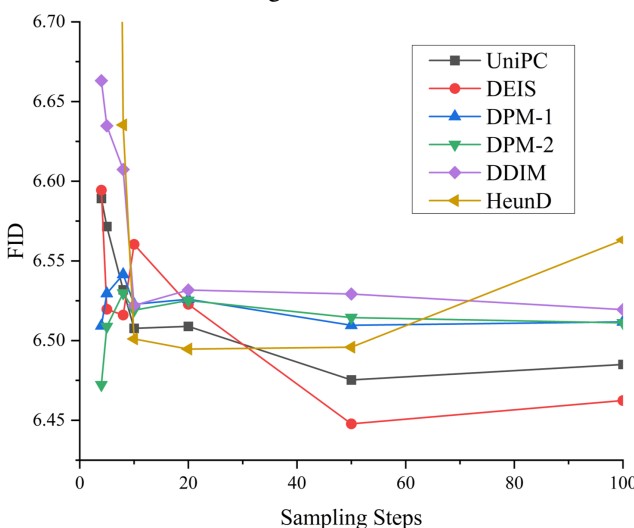

Figure 6: ILD w/o IA-VAE

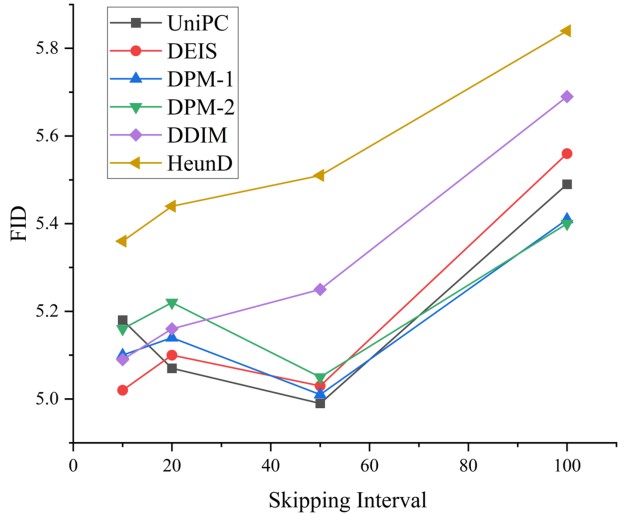

Figure 7: FILD

# E IMPLEMENTATION DETAILS

For pretraining of the ILD-VAE model, motion transformer encoders and decoders all consist of 11 layers and 8 heads with skip connection by default, the transformer-based denoiser of ILD is almost the same architecture as VAE, except for the 13 layers. We employ a frozen CLIP-ViT-L/14 model as the text encoder, yielding text embedding $c \in R^{768}$, and adopt the classifier-free guidance (Ho & Salimans, 2022) where the 10% random CLIP embeddings are set to zero during training and the guidance coefficient is set to 3 during sampling. The hyperparameters used in ILD are: $\lambda_{\text{MA}} = 0.01$, $\lambda_{\text{BL}} = 10$, $\lambda_{\text{FC}} = 30$, $\lambda_{\text{RO}} = 0.01$, $\lambda_{\text{DM}} = 3$, $\lambda_{\text{KL}} = 0.00001$, and $\overline{M} = 1$. All our models are trained with the AdamW optimizer using a fixed learning rate of 0.00001. Our mini-batch size is set to 128 during the whole training stage. The number of diffusion steps is 1000 during training, while 10 during inference using the DEIS sampling strategy (Zhang & Chen, 2022), and the variances are scaled linearly from 0.000085 to 0.012. For the FILD, the student network shares a similar architecture with ILD, and the $\lambda_{DMD}$ is set as 0.001, the skipping interval for the consistency model is 50, and we use UniPC solver (Zhao et al., 2023) as the ODE solver for the full chain reverse process. The training was performed on one A100 GPU, with ILD-VAE training taking 72 hours over 6000 epochs, ILD training taking 16 hours over 2000 epochs, and FILD training taking 20 hours over 2000 epochs. Testing was conducted on one A100 GPU.

**Pretained RQ-VAE Implementation Details.** We follow almost the same setting from Momask (Guo et al., 2024), except that we train it on InterHuman (Ionescu et al., 2013) and two individuals are encoded jointly into a single latent space. The batch size is set to 256. The learning rate reaches 2e-4 after 2000 iterations with a linear warm-up schedule. We employ 4-layer resblocks for both the encoder and decoder, with a down-scale factor of 4. The quantization dropout ratio $q$ is set to 0.2. The RVQ consists of 6 quantization layers, each with a codebook containing 512 codes with 512 dimensions. .

**Baseline setting.** We compare with various text-to-motion methods in two-person interactive scenarios, including single-person methods VAE-based TEMOS (Petrovich et al., 2022) and T2M (Guo et al., 2022), diffusion-based MDM (Tevet et al., 2022b), and the two-person diffusion-based method ComMDM (Shafir et al., 2023), InterGen (Liang et al., 2024), in2IN (Ponce et al., 2024), TIM (Wang et al., 2025), and InterMask (Javed et al., 2024) based on masked transformer. To conduct fair comparisons, the above single-person methods are trained with the same InterHuman and Inter-X training set and test set. To extend single-person motion synthesis models to handle two-person interaction, the networks' input and output dimensions are modified to accommodate the non-canonical representation of the InterHuman dataset. Specifically, we report the results of TIM, which was built upon a Transformer backbone.

# F EVALUATION METRICS

## F.1 FRECHET INCEPTION DISTANCE (FID)

The FID (Heusel et al., 2017) measures the distribution distance between the generated and real interaction features.

$$\text{FID} = \|\mu_{\text{gt}} - \mu_{\text{pred}}\|^2 - \text{Tr}(\Sigma_{\text{gt}} + \Sigma_{\text{pred}} - 2(\Sigma_{\text{gt}}\Sigma_{\text{pred}})^{\frac{1}{2}})$$

where $\mu_{\text{gt}}$ and $\mu_{\text{pred}}$ are the mean ground-truth and generated interaction features, and $\Sigma$ represents the covariance matrix.

## F.2 MULTIMODAL DISTANCE (MM-DIST)

This metric calculates the average Euclidean distances between each text feature and the generated interaction feature.

$$\text{MM-Dist} = \frac{1}{N} \sum_{i=1}^{N} \|f_{t,i} - f_{m,i}\|$$

where $f_{t,i}$ and $f_{m,i}$ are the features of the $i$th text-interaction pair.

### F.3 DIVERSITY

All generated interactions are randomly sampled to calculate the average Euclidean distances between two subsets.

$$\text{Diversity} = \frac{1}{X_d} \sum_{i=1}^{X_d} \|x_i - x_i'\|$$

### F.4 MULTIMODALITY (MMODALITY)

This metric assesses the variability given multiple text descriptions by calculating the average pairwise Euclidean distance between motion features.

$$\text{MModality} = \frac{1}{J_m \times X_m} \sum_{j=1}^{J_m} \sum_{i=1}^{X_m} \|x_{j,i} - x_{j,i}'\|$$

where $x_{j,i}$ and $x_{j,i}'$ are the features of the $j$th pair of the $i$th text description.

### F.5 INTERPENETRATION (COLLISION) METRIC

Our interpenetration metric is based on a Mesh-derived Signed Distance Field (SDF) (Jiang et al., 2020), which quantifies the distance of a point to a mesh surface, indicating whether the point is inside or outside. Specifically, for each human mesh $\mathcal{M}$, we first construct an explicit SDF field $\Phi_{\mathcal{M}}(p)$ by voxelizing the mesh and computing the signed distance for each voxel. This field provides continuous distance values for any query point $p$ via trilinear interpolation. The sign convention is negative for points inside $\mathcal{M}$ and positive for points outside.

For a query point $q$ with respect to a mesh $\mathcal{M}$, the SDF value $f_{\mathcal{M}}(q)$ is thus directly obtained from the precomputed field $\Phi_{\mathcal{M}}(q)$:

$$f_{\mathcal{M}}(q) = \Phi_{\mathcal{M}}(q) \tag{16}$$

For two interacting human meshes, $\mathcal{M}_A$ and $\mathcal{M}_B$, the interpenetration score is calculated symmetrically. We measure the extent to which mesh $\mathcal{M}_A$ penetrates $\mathcal{M}_B$, and vice versa, by summing the penetration depths of each mesh's vertices with respect to the other's SDF. The total collision is given by:

$$\mathcal{C}_{\text{inter}}(\mathcal{M}_A, \mathcal{M}_B) = \sum_{v_A \in \mathcal{M}_A} \sigma(-f_{\mathcal{M}_B}(v_A)) \cdot \mathbb{I}_{f_{\mathcal{M}_B}(v_A)<0} + \sum_{v_B \in \mathcal{M}_B} \sigma(-f_{\mathcal{M}_A}(v_B)) \cdot \mathbb{I}_{f_{\mathcal{M}_A}(v_B)<0} \tag{17}$$

where $v_A$ and $v_B$ are the sets of vertices for meshes $\mathcal{M}_A$ and $\mathcal{M}_B$, respectively. $\sigma(\cdot)$ is the sigmoid function, and $\mathbb{I}_{(\cdot)}$ is the indicator function. This formulation penalizes only points that are inside another mesh (i.e., $f < 0$), with the sigmoid function providing a smooth penalty based on penetration depth. A higher score indicates a more severe interpenetration between the two individuals.

## G HYPERPARAMETER ABLATION STUDY

Here, we conduct two different text-to-motion experiments on InterHuman dataset, which aims to explore the influence of hyperparameters in classifier-free diffusion guidance (Ho & Salimans, 2022). The first experiment is to change the text dropout $p$ from 0.1 to 0.5 while keeping the scale $s$ as 3.0. The second experiment changes the scale $s$ from 1.5 to 5.0 while keeping the text dropout $p$ at 0.1. In Tab. 4, we find that by changing dropout p from 0.1 to 0.2, the FID metric worsened, but the R Precision metric improved. And with the text dropout $p$ increase, all the fidelity metrics declined, with the diversity metric achieving its best result at $p = 0.3$. We assume that the higher the value of text dropout, the less information is available for text embedding, thus degrading performance. Furthermore, results indicate that as the guidance scale $s$ increases, both the FID and diversity metrics improve when the guidance is approximately 3.0. Meanwhile, the R Precision and Multimodal Distance metrics show improvement with higher guidance scales.

As shown in Tab. 5, we study the influence of batch size on model performance. Firstly, we increased the batch size from 32 to 512 while keeping the learning rate at 1e-4. As we can see, the batch size achieves the best results when set to 128, but the fidelity metrics are worse when the batch size is larger. Notably, with a batch size of 128, the GPU memory consumption is under 10 GB, allowing efficient training even on a single 2080ti GPU.

Table 4: Ablation study on the classifier-free guidance hyperparameters: text dropout probability $p$ and guidance scale $s$. '*' means we choose it for the final evaluation. The best performance for each metric is highlighted in bold.

| Methods | Classifier-free Dropout | Scale | R Precision top 1 ↑ | R Precision top 2 ↑ | R Precision top 3 ↑ | FID ↓ | Multimodal Dist ↓ | Diversity → |
|---|---|---|---|---|---|---|---|---|
| Real | - | - | $0.452^{\pm.008}$ | $0.610^{\pm.009}$ | $0.701^{\pm.008}$ | $0.273^{\pm.007}$ | $3.755^{\pm.008}$ | $7.948^{\pm.064}$ |
| ILD | $p=0.10$ | $s=3.0$ | $0.471^{\pm.007}$ | $0.615^{\pm.007}$ | $0.694^{\pm.007}$ | $4.935^{\pm.069}$ | $3.784^{\pm.002}$ | $7.965^{\pm.031}$ |
| ILD | $p=0.20$ | $s=3.0$ | $\mathbf{0.501^{\pm.006}}$ | $\mathbf{0.638^{\pm.004}}$ | $\mathbf{0.715^{\pm.005}}$ | $5.218^{\pm.064}$ | $3.782^{\pm.001}$ | $7.891^{\pm.029}$ |
| ILD | $p=0.30$ | $s=3.0$ | $0.493^{\pm.005}$ | $0.628^{\pm.004}$ | $0.698^{\pm.005}$ | $5.372^{\pm.074}$ | $3.789^{\pm.001}$ | $7.885^{\pm.031}$ |
| ILD | $p=0.40$ | $s=3.0$ | $0.475^{\pm.005}$ | $0.609^{\pm.006}$ | $0.686^{\pm.005}$ | $5.516^{\pm.062}$ | $3.797^{\pm.002}$ | $\mathbf{7.938^{\pm.032}}$ |
| ILD | $p=0.50$ | $s=3.0$ | $0.478^{\pm.006}$ | $0.612^{\pm.004}$ | $0.690^{\pm.004}$ | $5.541^{\pm.076}$ | $3.795^{\pm.001}$ | $7.842^{\pm.037}$ |
| ILD | $p=0.10$ | $s=1.5$ | $0.412^{\pm.005}$ | $0.551^{\pm.006}$ | $0.635^{\pm.006}$ | $6.251^{\pm.095}$ | $3.815^{\pm.001}$ | $7.674^{\pm.029}$ |
| ILD | $p=0.10$ | $s=2.0$ | $0.449^{\pm.006}$ | $0.589^{\pm.006}$ | $0.672^{\pm.006}$ | $5.133^{\pm.078}$ | $3.801^{\pm.002}$ | $7.795^{\pm.031}$ |
| ILD | $p=0.10$ | $s=2.5$ | $0.478^{\pm.005}$ | $0.614^{\pm.006}$ | $0.691^{\pm.007}$ | $4.887^{\pm.071}$ | $3.792^{\pm.002}$ | $7.869^{\pm.031}$ |
| ILD* | $p=0.10$ | $s=3.0$ | $0.495^{\pm.005}$ | $0.630^{\pm.005}$ | $0.709^{\pm.004}$ | $\mathbf{4.869^{\pm.073}}$ | $\mathbf{3.777^{\pm.001}}$ | $7.976^{\pm.027}$ |
| ILD | $p=0.10$ | $s=3.5$ | $0.482^{\pm.007}$ | $0.618^{\pm.007}$ | $0.697^{\pm.007}$ | $4.976^{\pm.072}$ | $3.783^{\pm.001}$ | $7.995^{\pm.031}$ |
| ILD | $p=0.10$ | $s=4.0$ | $0.484^{\pm.007}$ | $0.620^{\pm.007}$ | $0.700^{\pm.007}$ | $5.102^{\pm.075}$ | $3.782^{\pm.001}$ | $8.063^{\pm.030}$ |
| ILD | $p=0.10$ | $s=4.5$ | $0.486^{\pm.007}$ | $0.619^{\pm.007}$ | $0.701^{\pm.007}$ | $5.245^{\pm.079}$ | $3.781^{\pm.001}$ | $8.079^{\pm.030}$ |
| ILD | $p=0.10$ | $s=5.0$ | $0.487^{\pm.006}$ | $0.620^{\pm.007}$ | $0.699^{\pm.006}$ | $5.413^{\pm.082}$ | $3.782^{\pm.001}$ | $8.088^{\pm.029}$ |

As illustrated in Tab. 6, we present a more detailed comparison of the interactive and geometric loss designs. Overall, the interactive loss contributes more than the geometric loss. The RO loss function seems to be the most important among the various loss functions. Without it, the FID metrics get the worst results, and the diversity metric achieves an abnormally high value, highlighting the significance of the relative root orientation information. In contrast, the model gets minimal impact without the foot contact loss function, and interestingly, the R precision exhibits slight improvement. This suggests that the limited capacity of the latent space may hinder the effective learning of meaningful but challenging-to-contain foot contact status information. Since the similar significance of various metrics, we aim to select parameters for high-quality interaction generation, by primarily focusing on FID while considering R-precision as a secondary target.

Table 5: Ablation study on the batch size. Our final model, ILD, achieves the best trade-off in performance and stability with a batch size of 128, which is used for all other experiments. '*' indicates the chosen configuration.

| Methods | Batch Size | R Precision top 1 ↑ | R Precision top 2 ↑ | R Precision top 3 ↑ | FID ↓ | Multimodal Dist ↓ | Diversity → |
|---|---|---|---|---|---|---|---|
| Real | - | $0.452^{\pm.008}$ | $0.610^{\pm.009}$ | $0.701^{\pm.008}$ | $0.273^{\pm.007}$ | $3.755^{\pm.008}$ | $7.948^{\pm.064}$ |
| ILD | 32 | $0.481^{\pm.006}$ | $0.628^{\pm.006}$ | $0.701^{\pm.006}$ | $5.952^{\pm.092}$ | $3.784^{\pm.001}$ | $7.981^{\pm.037}$ |
| ILD | 64 | $\mathbf{0.499^{\pm.005}}$ | $\mathbf{0.640^{\pm.005}}$ | $\mathbf{0.713^{\pm.004}}$ | $5.315^{\pm.079}$ | $3.780^{\pm.001}$ | $7.915^{\pm.036}$ |
| ILD* | 128 | $0.495^{\pm.005}$ | $0.630^{\pm.005}$ | $0.709^{\pm.004}$ | $\mathbf{4.869^{\pm.073}}$ | $\mathbf{3.777^{\pm.001}}$ | $\mathbf{7.976^{\pm.027}}$ |
| ILD | 256 | $0.235^{\pm.005}$ | $0.351^{\pm.005}$ | $0.428^{\pm.006}$ | $10.104^{\pm.099}$ | $3.901^{\pm.002}$ | $7.896^{\pm.033}$ |
| ILD | 512 | $0.128^{\pm.002}$ | $0.203^{\pm.004}$ | $0.261^{\pm.004}$ | $9.897^{\pm.118}$ | $3.972^{\pm.002}$ | $7.781^{\pm.035}$ |

Table 6: We study the influence of the loss function and model architecture on text-to-motion. '*' means we choose it for the final evaluation.

| Methods | R Precision top 1 ↑ | R Precision top 2 ↑ | R Precision top 3 ↑ | FID ↓ | Multimodal Dist ↓ | Diversity → |
|---|---|---|---|---|---|---|
| Real | $0.452^{\pm.008}$ | $0.610^{\pm.009}$ | $0.701^{\pm.008}$ | $0.273^{\pm.007}$ | $3.755^{\pm.008}$ | $7.948^{\pm.064}$ |
| ILD w/o DM Loss | $0.469^{\pm.005}$ | $0.618^{\pm.007}$ | $0.695^{\pm.005}$ | $5.201^{\pm.074}$ | $3.765^{\pm.001}$ | $7.859^{\pm.030}$ |
| ILD w/o RO Loss | $0.455^{\pm.005}$ | $0.615^{\pm.009}$ | $0.683^{\pm.008}$ | $5.738^{\pm.067}$ | $3.841^{\pm.002}$ | $8.102^{\pm.028}$ |
| ILD w/o Interactive Loss | $0.451^{\pm.006}$ | $0.602^{\pm.006}$ | $0.671^{\pm.005}$ | $6.145^{\pm.108}$ | $3.795^{\pm.002}$ | $7.996^{\pm.020}$ |
| ILD w/o BL Loss | $0.478^{\pm.005}$ | $0.614^{\pm.005}$ | $0.690^{\pm.005}$ | $5.072^{\pm.067}$ | $3.832^{\pm.001}$ | $7.785^{\pm.033}$ |
| ILD w/o FC Loss | $0.471^{\pm.003}$ | $0.625^{\pm.005}$ | $0.700^{\pm.004}$ | $5.013^{\pm.061}$ | $3.839^{\pm.001}$ | $7.882^{\pm.036}$ |
| ILD w/o Geometric Loss | $0.467^{\pm.004}$ | $0.619^{\pm.004}$ | $0.680^{\pm.005}$ | $5.189^{\pm.071}$ | $3.788^{\pm.001}$ | $7.924^{\pm.036}$ |

## H    COMPUTATIONAL RESOURCE

Table 7: Comparative analysis of computational efficiency and performance. Metrics are calculated on an A100 GPU.

| Method | Params (M) | FLOPs (G) | AITS (s) | FID | FPS |
|---|---|---|---|---|---|
| InterGen | 182.2 | 80.5 | 2.89 | 5.918 | <1 |
| in2IN | 184.8 | 87.4 | 2.98 | 5.177 | <1 |
| InterMask | 126.5 | 43.9 | 0.77 | 5.154 | 1 |
| ILD | 38.4 | 22.3 | 0.09 | 4.869 | 10 |
| FILD | 38.4 | 7.8 | 0.03 | 4.980 | 30 |

## I    THE USE OF LARGE LANGUAGE MODELS (LLMS)

In preparing this manuscript, Large Language Models (LLMs) were utilized as a writing assistance tool. Their use was strictly limited to improving the language and readability of the text, including correcting grammar and refining sentence structure. The LLMs played no role in the research ideation, data analysis, or the formulation of scientific conclusions. The authors take full responsibility for all content presented.

