# OpenReview forum: "SHAPING LATENT DIFFUSION FOR EFFICIENT TEXT-CONDITIONED INTERACTION GENERATION"
_ICLR.cc/2026/Conference — ICLR 2026 Conference Withdrawn Submission_

### Official Review · Reviewer_35Kf · 2025-10-25

**Soundness:** 3
**Presentation:** 3
**Contribution:** 3
**Rating:** 6
**Confidence:** 4

**Summary:**

The paper proposes Interaction Latent Diffusion (ILD) with an interaction-aware, multi-token latent space aligned to a pretrained motion tokenizer, plus a second-order DEIS sampler (10 steps) and a lightweight neural collision guidance. It then distills ILD into a one-step real-time model, FILD. On InterHuman, ILD reports FID 4.869, and FILD boosts inference from ~10 FPS to ~30 FPS.

**Strengths:**

1.FILD uses consistency + distribution-matching distillation for robust one-step generation.
2.Interaction-aware multi-token latent space with RVQ-VAE alignment improves capacity without heavy extra cost.
3.Thorough evaluation/ablations on InterHuman/Inter-X (incl. solver, collision, and component studies).

**Weaknesses:**

1.Diversity/multimodality leaves room for improvement; authors hypothesize freezing IA-VAE may hurt diversity.
2.Tokenizer alignment depends on a pretrained RVQ-VAE and linear projection; robustness under domain/skeleton shifts is unclear.
3.Physics is test-time guided; limited training-time physical inductive bias may cap realism.

**Questions:**

1.Still see the foot slides in the video.
2.Could adding training-time physics priors (e.g., implicit SDF supervision or contact replay) further improve diversity and plausibility?

---

### Official Review · Reviewer_hxL3 · 2025-10-31

**Soundness:** 3
**Presentation:** 2
**Contribution:** 2
**Rating:** 2
**Confidence:** 4

**Summary:**

This paper proposes a diffusion-based framework for interaction generation. Within the classification diffusion paradigm for motion generation, the authors revise the architecture by introducing three key components: (1) a discrete token alignment mechanism in the VAE, (2) a fast neural collision guidance module to update the latent code during latent denoising, and (3) a distillation technique for accelerated generation. Experiments are conducted on two large-scale datasets to evaluate the proposed approach.

**Strengths:**

1. The proposes method includes various mechanisms to ensure synthesis fidelity. Glad to see that the collective engineering integration works.
2. Extensive quantitative experiments are performed to demonstrate the effectiveness of the proposed components, showing consistent improvements across various metrics.

**Weaknesses:**

1. The proposed method modifies multiple components within the classical diffusion framework for motion generation. While these engineering refinements collectively improve performance, the theoretical novelty appears limited, as many of the introduced techniques have been explored in prior works. Overall, the paper presents a solid integration of existing methods for enhanced performance but lacks a clear conceptual takeaway or deeper insight.
2. The dimensionality trade-off solution raises two concerns. The method leverages a pre-trained RVQ-VAE as an alignment target: (a) Since the RVQ-VAE is trained for single-human motion generation, would it introduce bias or alignment errors when applied to multi-agent interaction scenarios? (b) If the RVQ-VAE demonstrates strong performance, why not directly replace the IA-VAE with it? What specific advantages does the proposed alignment design offer compared to simply adopting RVQ-VAE?
3. Formatting issue: The manuscript formatting appears inconsistent to ICLR template, particularly in the spacing before each section. It is recommended to standardize the formatting for readability and compliance with publication guidelines.

**Questions:**

The dimensionality trade-off solution raises two concerns. The method leverages a pre-trained RVQ-VAE as an alignment target: (a) Since the RVQ-VAE is trained for single-human motion generation, would it introduce bias or alignment errors when applied to multi-agent interaction scenarios? (b) If the RVQ-VAE demonstrates strong performance, why not directly replace the IA-VAE with it? What specific advantages does the proposed alignment design offer compared to simply adopting RVQ-VAE?

---

### Official Review · Reviewer_kgAA · 2025-10-31

**Soundness:** 2
**Presentation:** 1
**Contribution:** 1
**Rating:** 2
**Confidence:** 5

**Summary:**

This paper introduces the Interaction Latent Diffusion (ILD) model, a novel method for generating realistic, text-conditioned multi-person human motions. ILD addresses the limitations of previous models by using an interaction-aware, multi-token latent space that better captures the complex spatio-temporal dynamics between individuals. To improve efficiency and physical realism, the model incorporates a lightweight neural collision guidance system and a high-order ODE solver, enabling high-fidelity generation in just 10 steps. Furthermore, the authors present Flash ILD (FILD), a distilled version of the model that utilizes a tailored consistency distillation pipeline to achieve real-time, one-step generation at 30 FPS while maintaining high quality.

**Strengths:**

The proposed ILD model achieves good performance in generation quality.

The paper successfully tackles the critical challenge of inference speed by developing Flash ILD (FILD), a distilled model capable of accelerating generation to real-time speeds of 30 FPS, an improvement over the base model's 10 FPS.

**Weaknesses:**

The contribution of this work is limited. The technical part mainly extends existing methods, like MotionLCM.

The comparison in Fig. 1 misses the MotionLCM and its V2 version.

The experiment misses reporting the results on different inference steps, like 100 steps.

AIST should also be reported in Tab. 1 with baselines and MotionLCM-v2.

**Questions:**

see above

**Details Of Ethics Concerns:**

This paper adjusts the space between sections and subsections. See the title in sec. 2/3/4. Shall I continue reviewing this paper? Is this a violation of the submission guidelines and needs to be desk rejected?

---

### Official Review · Reviewer_RZCY · 2025-10-31

**Soundness:** 3
**Presentation:** 3
**Contribution:** 2
**Rating:** 4
**Confidence:** 4

**Summary:**

This paper introduces Interaction Latent Diffusion (ILD) and its distilled variant Flash ILD (FILD) for text-conditioned multi-person motion generation. Unlike single-token latent diffusion models, ILD uses a multi-token, interaction-aware VAE with geometric and interactive constraints (e.g., joint distance and orientation losses) and aligns its latent space with a pretrained residual VQ-VAE tokenizer to better capture interpersonal dynamics. Efficiency and physical realism are enhanced through a neural collision guidance module and a second-order ODE solver (DEIS) that reduces denoising steps to 10.

**Strengths:**

- ILD systematically addresses limitations of single-token latent diffusion by designing an interaction-aware, multi-token latent space with explicit geometric and interpersonal regularization.

- Combining a second-order DEIS solver and VolumetricSMPL-based neural collision guidance enables physically plausible interactions.

**Weaknesses:**

1. While ILD achieves the best FID (4.869 vs 5.154 for InterMask) and slightly higher R-Precision (0.495 vs 0.449), the gains are relatively small and within reported variance ranges. It is difficult to isolate how much improvement stems from the new interaction-aware losses versus architectural scaling or alignment with a pretrained tokenizer. A clearer ablation isolating each major factor (latent alignment vs. interactive loss vs. DEIS solver) on identical diffusion setups would strengthen causal interpretation.

2. The paper includes only a few visual examples (Fig. 3) contrasting ILD and InterMask. Given the modest numerical differences, more qualitative comparisons or user studies would help substantiate the claim of higher physical realism and interaction quality. Including side-by-side videos with synchronized prompts (e.g., “hand-over,” “helping up”) across various methods would make the improvement more evident.

3. The FILD training pipeline (consistency + distribution matching) is intricate (Eqs. 7–9). However, the paper lacks analysis of failure modes or stability trade-offs in distillation—e.g., how removing DMD or modifying guidance coefficients affects convergence. Adding training stability metrics or visualizing student–teacher divergence curves would clarify why FILD maintains fidelity under one-step generation.

4. ILD is evaluated only on two-person interactions. It is unclear how well the model scales to more participants or to longer sequences with diverse contact patterns. A short discussion or experiment (even qualitative) on three-person or crowd scenarios would broaden the paper’s impact and demonstrate generalizability.

5. It is suggested to include more recent SoTA methods for comparison, such as [a] and [b].

    [a] Tlcontrol: Trajectory and language control for human motion synthesis

    [b] MaskControl: Spatio-Temporal Control for Masked Motion Synthesis

Minor issues:

- Figure 2 could better annotate “Stage 1–3” transitions (IA-VAE → ILD → FILD) and denote latent dimensions.

- Equations (5–6) describing the DEIS solver are mathematically dense; summarizing its intuition (why DEIS works better for 10-step sampling) in the main text would help accessibility.

- The authors seem to change the margins of different sections and paragraphs to squeeze in more content. However, this sometimes reduces the readability.

**Questions:**

Please refer to the weakness section.

---

### Note · Authors · 2025-11-12

I have read and agree with the venue's withdrawal policy on behalf of myself and my co-authors.